# Syngas Production and Combined Heat and Power from Scottish Agricultural Waste Gasification—A Computational Study

Ahmed M. Salem [1,2] , Harnek S. Dhami [2] and Manosh C. Paul [2,*]

1    Mechanical Power Department, Faculty of Engineering, Tanta University, Tanta 31521, Egypt;
     ahmed_salem@f-eng.tanta.edu.eg
2    Systems, Power & Energy Research Division, James Watt School of Engineering, University of Glasgow,
     Glasgow G12 8QQ, UK; harrydhami@hotmail.co.uk
*    Correspondence: manosh.paul@glasgow.ac.uk; Tel.: +44-(0)141-330-8466

**Abstract:** This paper explores the possibility of utilizing Scottish agricultural waste for sustainable energy, including combined heat and power (CHP). Challenges of using unconventional agricultural feedstocks for gasification are addressed, and the study is focused on the fundamental understanding of gasification processes as well as the design constraints of a commonly used downdraft gasifier. An integrated kinetic and CHP model is presented to address these, and the results demonstrate the optimal working parameters that maximize the production of high-quality syngas and power from the CHP engine. Based on the robust sensitivity analysis, an equivalence ratio ($\Phi$) of 0.3–0.35 with moisture content (MC) less than 10% yields higher production of syngas, thus resulting in higher gasification efficiency. Moreover, an increase in $\Phi$ also favors the gasification temperature, which promotes tar cracking and results in lower tar content. Additionally, the gasification efficiency, design limitations, and challenges are addressed to optimize the gasifier design so that it can handle diverse feedstocks with high performance. Therefore, the findings are significant in the field of bioenergy and, in particular, help to expand the route of converting agricultural waste to energy.

**Keywords:** agricultural feedstocks; waste; gasification; syngas; CHP; kinetic modeling; bioenergy

## 1. Introduction

The world's population is expected to increase to 10 billion by 2050 [1], and the resulting energy needs will expand by 56% [2]. As a result, the challenges are rising to reduce the fossil fuels dependency and the use of renewables and other sources such as waste. The forest and agricultural residues are available almost everywhere. Besides the wide diversity, it is a renewable source of energy that is also able to reduce greenhouse gas emissions (GHG) [3]. The GHG produced over the last century from anthropological activities is estimated as the largest driver of the change in climate [2]. As the world population is increasing rapidly, our basic need for energy, water, and food is also increasing accordingly.

Gasification is a remarkable technology that is used to produce energy from different sources, e.g., waste, biomass, agricultural and industrial residues. It also plays a vital role in reducing GHG production globally [4]. Another form for generating energy from waste with the ability to utilize the maximum amount of heat generated is combined heat and power (CHP). The CHP process produces power, and during the process of power generation, it utilizes the heat generated in different sources, e.g., boilers and heaters. Conventional sources of electricity production produce heat, which is normally wasted through the atmosphere. Large power stations contain cooling towers that dissipate the excess heat, resulting in lower efficiencies of such a plant compared to modern CHP plants. Typical gas/coal-fired power plants have efficiencies of 40–50% compared to 80% efficiencies of CHP plants, which use wasted heat [5]. CHP plant exports heat or even cooling powers in factories and local buildings for industrial and residential purposes. The

electricity produced from conventional power plants is distributed over the national grid with losses in energy/power due to transmission and distribution (~7–9%) [6]. On the other hand, CHP is not expecting losses in energy transfer because the power is locally generated, and the waste heat is usually recovered (Figure 1). Consequently, the technology is more advantageous and has higher efficiencies in electricity and heat distribution [7].

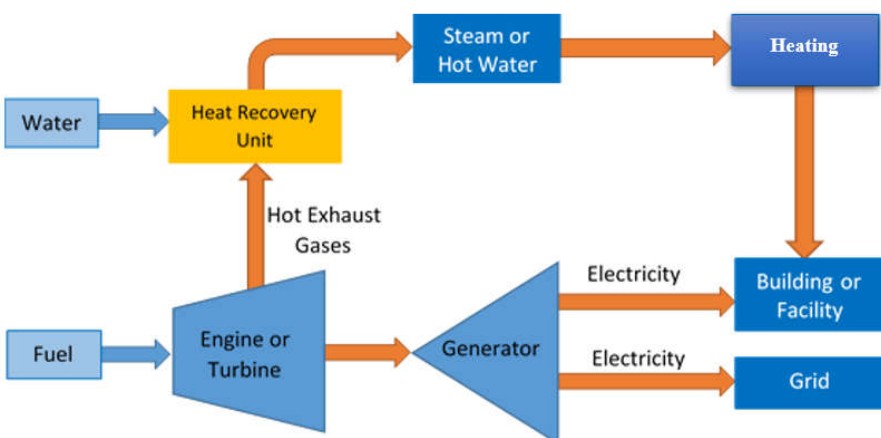

**Figure 1.** Process of a combustion-based CHP System.

Although CHP energy is derived from different engines, the current research focuses on the use of gasifiers. Gasification and CHP power generation have a wide potential to generate thermal and electric powers [8,9]. More precisely, the produced gas from downdraft biomass gasifier is further combusted in internal combustion engines to generate electricity and heat energy. Simultaneously, the gasifier produces heat that could be recovered for further applications (e.g., hot water and boilers). CHP with biomass gasifiers has been used widely in many applications [10–13]. Most of these applications rely on energy production and regenerating heat using biomass, coal, and combined cycles using biomass and natural gas or other gases for CHP applications. Some other recent researchers rely on using waste as feeding for gasifiers [14,15]. Agriculture residues considered in this study are valuable feedstock for gasification and CHP applications.

A downdraft gasifier in the current research is modeled using the kinetic code recently developed by the authors [4] for the optimization of the downdraft biomass gasifier's work and performance. The model is combined with a CHP engine for predicting the overall energy production. As already evidenced in the published literature, a downdraft gasification model can be developed using an equilibrium [16,17], thermochemical kinetics [4,18,19], and computational fluid dynamics (CFD) approach [20–22]. However, as also illustrated in those studies, a pure thermodynamic equilibrium model is not sufficiently able to predict the product gases of a gasifier because it overpredicts the $H_2$ production and corresponding heating value. The equilibrium model also predicts a lower amount of CO with higher amounts of $CH_4$ compared to those of the experimental studies [23], whereas a kinetic model gives much more accurate gas composition prediction with the estimation of the gasification temperature as well as velocity along a gasifier. Nevertheless, the existing kinetic model [4] may be sensitive to the chemical contents presented in a biomass feedstock, and in some cases, it might even overpredict the gasification outputs. Thus, the paper first tries to improve the kinetic model and link it to a CHP engine model aiming to investigate the gasification process of Scottish agricultures waste to study such feedstocks' feasibility for gasification. The model has been widely validated over a wide range of materials [4]. However, the current materials have a different range of CHO. As a result, the current model aims to address this challenge and then make the model applicable for a wide range of feedstocks. Through an in-depth sensitivity analysis of the feedstocks, the study is also focused on the determination of the gasifier optimum working parameters

leading to higher syngas quality and production. Furthermore, the model addresses the issues of gasifier design and challenges of multiple feedstocks utilization

Regarding the availability of the feedstocks analyzed for the purposes of this research, in Scotland, an estimated amount of 462,000 hectares of cereals and oil seeds was grown in 2016 [24]. The barley is the main crop in Scotland, where around 286,000 and 110,000 hectares of barley and wheat, respectively, were grown in 2016. Moreover, 31,000 and 30,000 hectares of oats and oilseed rape are grown, respectively. Barley is sown in spring and autumn, whereas around 80% of the Scottish crops are spring barley and sown between March and April [25]. The straw that could be used as biomass feedstocks is estimated at 2.75–4 tons/year from UK farms, in which it is mostly used as fertilizer, livestock beddings, or animal feed. However, it also could be used as a biomass gasification feeder because of considerable amounts of C, H, and O. The ultimate and proximate analysis data of such crops show similar values to typical biomass. In the past, farmers used to burn waste from agricultural residues [26]. However, from 2019, this is not allowed anymore due to GHG emissions, and as a result, new recycling pathways are considerable. The report published by [27] estimates that energy production from Scottish waste could meet 31% of the renewable energy heat target and up to 4.3% of the electricity grid. As a result, such feedstocks have the advantage of being gasified and further integrated with CHP for energy production, and this study, for the first time, explores this potential opportunity.

Based on the review carried out, the current research novelty is aiming to study such feedstocks feasibility for gasification, including the gasification efficiency and second law analysis (i.e., exergy efficiency), comparing this with biomass materials. Additionally, the current work aims to optimize the process gasification for agricultural residues by carrying out a detailed sensitivity analysis for maximizing syngas output, heating value, and decreasing the amount of tar produced. Furthermore, there is a wide variety of materials compositions (ultimate and proximate analysis). As a result, a key challenge of the current research is to design a gasifier that is able to handle such diversity for maximizing the output of the gasification process. Furthermore, the research investigates the feed rates required to deliver thermal and electric outputs from the coupled CHP engine at any given load in addition to the gasifier design dimensions, gas concentrations, and tar content.

## 2. Methodology

Figure 1 illustrates a combustion-based CHP system powering a turbine or reciprocating engine. CHP systems are flexible and can operate on a wide scale of fuels such as natural gas, oils, or biogas. The fuel is combusted in a turbine combustor or an internal combustion engine (ICE). The engine or turbine turns the generator, which in turn generates electricity. The electricity powers local buildings, and excess electrical generation is exported to the national grid. The heat generated from combustion is captured by means of a heat recovery unit, and the steam or hot water produced from the heat recovery is exported to local buildings. The same principle used in Figure 1 is also applied for using CHP with biomass and waste to produce clean energy. Waste is going to be fed into the gasifier, while the producer gas is combusted in an ICE to produce power. The design optimization of a suitable gasifier for a required electric and thermal load depends on some factors such as the feedstock, MC, equivalence ratio, and the efficiency of the CHP engine.

The University of Glasgow recently completed an installation of a district heating scheme to meet the thermal and electrical power requirements of the campus. For this study, it was decided to incorporate the CHP engine currently in operation into the simulation model. A detailed technical model was developed to build up the design for the engine and its syngas production rate that could produce the same energy for running the University CHP system. The combined model is composed of two sub models. The first sub-model is the gasification model (kinetic) [4], while the other model is the CHP model. By combining both models, the physical dimensions of the gasifier required to power a CHP engine for the selected agricultural waste is determined. The combined model requires the thermal

power and the feedstock ultimate analysis data (Figure 2) to predict the gasifier behavior and outputs. Then, it predicts all of producer gas production, tar content, heating value, temperature along the gasifier, and the optimum gasifier design (dimensions), e.g., diameter, height of every zone, nozzles diameter, and throat diameter.

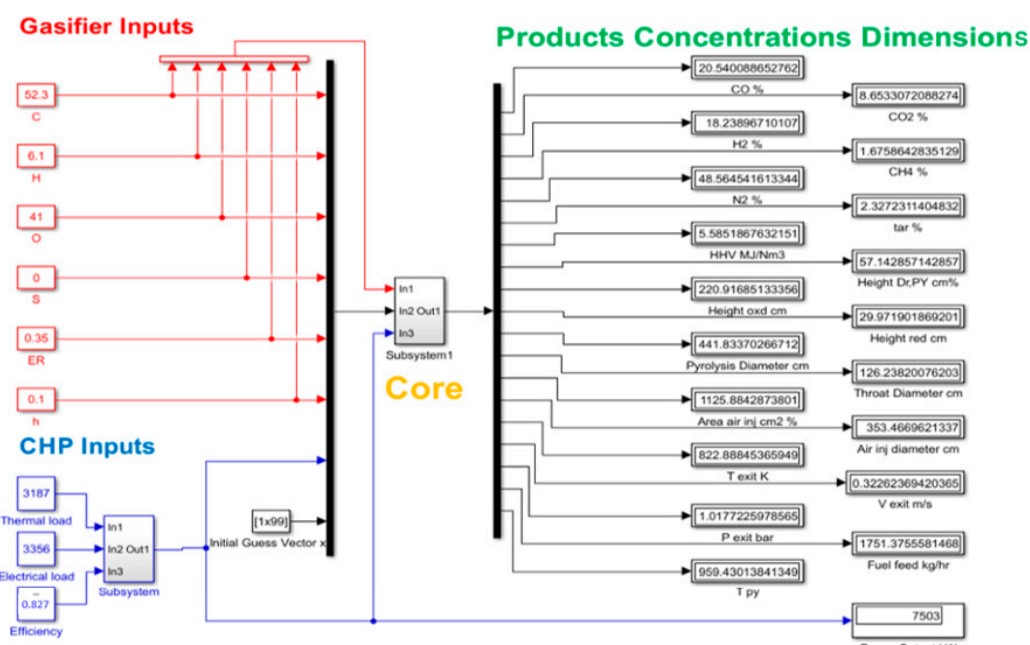

**Figure 2.** General model of gasifier and CHP combined.

### 2.1. Gasifier Model

Under Matlab environment, the kinetic model for a downdraft gasifier was built in by Salem and Paul [4], which incorporated detailed chemical kinetics for every zone within the gasifier, enabling an estimation of the working parameters of a downdraft gasifier. By entering the ultimate analysis, the model can predict the optimum gasifier design, tar content, and the concentration of producer gas. For the current research, the MATLAB code developed was converted to a user-friendly Simulink model, as shown in Figure 2, and then integrated with the CHP model also developed using Simulink. The kinetic model was previously validated against numerical and experimental data over a wide range of feedstocks under different working conditions and found a good agreement [4]. As a result, no further validation is stated in the current research.

### 2.2. CHP Model

The University of Glasgow CHP system [28,29] is considered as a benchmark operation system for the development of gasification CHP in this work. The engine is a 4-stroke IC engine, capable of delivering 7.5 MW at full load, 3.35 MW$_{el}$ and 3.25 MW$_{th}$ with a combined efficiency of 87.3%. The engine operates on natural gas, and since the gas flow rate required to power any IC CHP engine varies in accordance with the *LHV*, the *LHV* of syngas produced from the gasifier must be calculated.

Changing the power output of the gasifier has no effect on the producer gas concentration [4]. Therefore, in order to calculate the *LHV*, a simulation was run to determine the *LHV* for a gasifier base load of 20 kW, as illustrated in Figure 2. The lower heating value of syngas produced was calculated using Equation (1). The hydrogen, carbon monoxide, and methane values were taken from the simulation outputs.

$$LHV = 10.78 \, H_2 + 12.63 \, CO + 35.88 \, CH_4. \tag{1}$$

The required syngas flow rate for the engine was calculated using Equation (2).

$$\dot{m}_{syngas} = \frac{P}{LHV} \tag{2}$$

$$P = (Th \times \eta_{th}) + (El \times \eta_{el}) \tag{3}$$

where $P$ is power of CHP engine, $Th$ is thermal output of CHP engine, $El$ is electrical output of CHP engine, $\eta_{th}$ is thermal efficiency of CHP engine, and $\eta_{el}$ is electrical efficiency of the engine, while the biomass heating value is calculated from [30] as follows:

$$HHV_{solid\ fuel} = 0.519\,C + 1.625\,H + 0.001\,O^2 - 17.87. \tag{4}$$

The syngas flow rate of gasifier is less than the required gas flow rate of the CHP engine; therefore, an iterative loop was created to adjust the gasifier accordingly to meet the gas flow rate of the CHP engine. The kinetic chemical model found that changing the thermal power output of the gasifier results in a change in the physical gasifier dimensions, with no effect on the producer gas. The produced gas composition depends on the working parameters of (feedstock type, MC, and $\Phi$). If the gasifier dimensions changed (different feed rate of biomass), with the same feedstock type, MC, and $\Phi$, this means the producer gas composition will remain the same (i.e., same CV). The kinetic model further assumes one mol of the biomass and predicts the producer gas composition based on this. The thermal power of the gasifier is directly related to the gasifier fuel feed rate and throat diameter and thus the gas flow rate. It was reported in [4] that, as the power requirement of the gasifier is increased, the biomass feed rate must also increase. Therefore, to accommodate for a larger biomass feed rate, the throat diameter must increase, and subsequent impacts on the required CHP gas flow rate and the power of the gasifier must also need to be optimized accordingly. The producer gas $HHV$ (MJ/Nm$^3$) is calculated as follows:

$$HHV = 12.74\,H_2 + 12.63\,CO + 39.82\,CH_4. \tag{5}$$

## 3. Farm Waste as a Gasification Feedstock

Eight different agriculture waste feedstock samples were collected from Scottish Farms (Cragnathrow, Kinkell, and Foulis), as shown in Figure 3. The corresponding ultimate and proximate analyses of these feedstocks, performed at SGS United Kingdom Limited [31], are presented in Figure 4.

The values of CHO for all the feedstocks are quite similar to other biomass (wood chips), which makes it suitable for the process gasification. However, a higher MC level for spent barley and spent hops was found. As a result, for economic gasification, this requires further drying (<20% MC) before feeding to a gasifier.

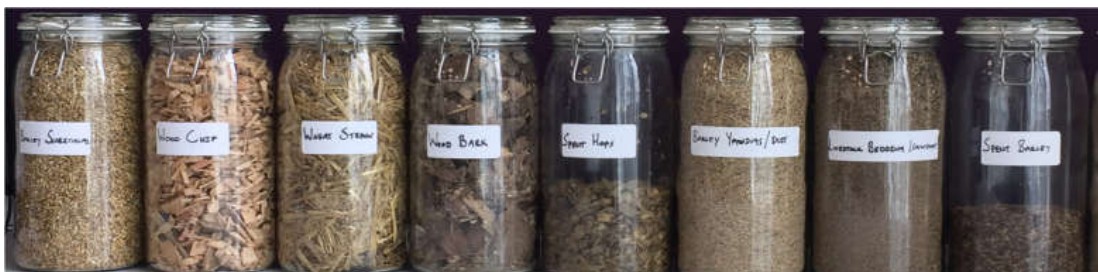

**Figure 3.** Samples of agriculture waste feedstock.

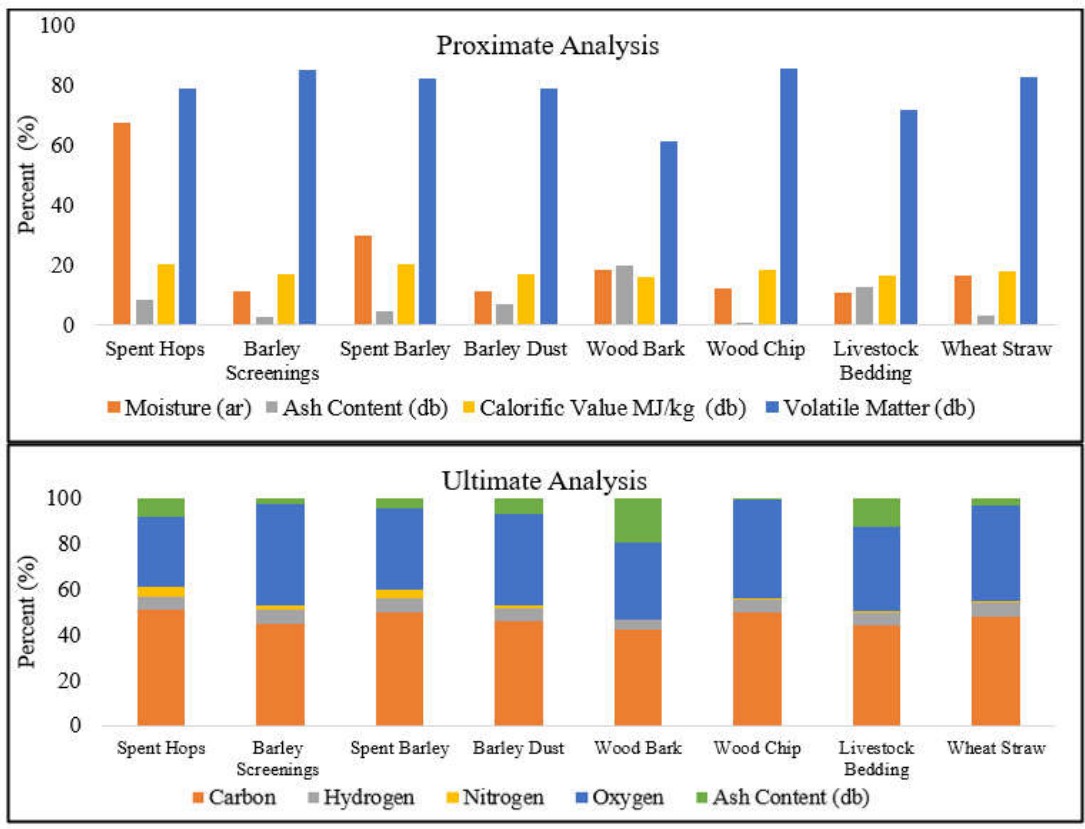

**Figure 4.** The ultimate and proximate analysis of the used feedstocks.

Table 1 illustrates typical results for the ultimate and proximate analysis for different conventional biomass materials. Comparing the current agricultural feedstocks with the biomass materials, they have similar properties and, hence, the ability to be gasified within a specific range. For example, the volatiles and fixed carbon (FC) content are important components in any gasification process. When both types of feedstocks are compared, they have values of volatiles around 80%. On the other hand, moisture content level, which affects the quality of the gasification process, has always been considered. Higher values of moisture require more energy for removal (spent hops and spent barley feedstocks). All the other feedstocks have lower MC levels <20%, which again lies in the normal range of typical biomass.

**Table 1.** Typical conventional biomass characteristics.

| | Ultimate Analysis db. % | | | | Proximate Analysis db. % | | | | Ref. |
|---|---|---|---|---|---|---|---|---|---|
| | **C** | **H** | **O** | **N** | **Volatiles** | **FC** | **Ash** | **MC** | |
| Rubber wood | 50.6 | 6.5 | 42 | 0.2 | 81.1 | 19.1 | 0.7 | 18.5 | [22] |
| Neem | 45.1 | 6.0 | 41.5 | 1.7 | 81.75 | 12.65 | 5.6 | 10 | [32] |
| Wood pellets | 50.7 | 6.9 | 42.4 | - | 80.5 | 18.5 | 0.9 | 6.07 | [33] |
| Bamboo | 48.39 | 5.86 | 39.21 | 2.04 | 80.3 | 15.2 | 4.5 | 10 | [32] |

The major input parameters and assumptions to the model are presented in Table 2. Any other parameters are derived directly from the model, e.g., gas composition, feeding mass flow rate, heating value, and gasifier dimensions.

**Table 2.** Boundary conditions, assumptions, and input parameters to the model.

| | |
|---|---|
| **Input Parameters** | Ultimate analysis CHOS<br>MC, $\Phi$, and required thermal power<br>Input T = 298 K<br>Input pressure = 1 atm |
| **Assumptions** | Char is fully consumed in reduction stage<br>Air is used as a gasifying agent<br>Steady-state model<br>Volatiles (CO, $CO_2$, $CH_4$, $H_2$, and $H_2O$)<br>Tar is assumed as a single compound ($C_6H_{6.2}O_{0.2}$) |

## 4. Results and Discussion

### 4.1. Syngas and CHP Production

The results in Figure 5 show the effect of the feeding rate of the gasifier on the power output at different loads. A gasifier design is accommodated based on the required thermal power, which is used as feed for the CHP engine.

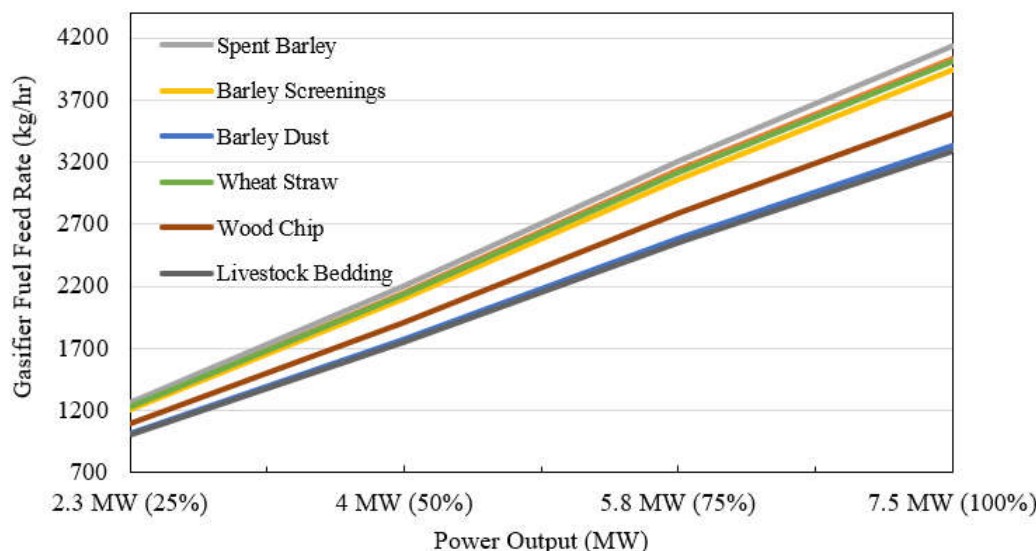

**Figure 5.** Fuel feed rate of gasifier/CHP vs. power output.

Generally, as seen in this figure, increasing the fuel feeding rate constantly increases the power output as per the relation presented in Equation (2). The spent barley sample, which requires the highest feeding rates to achieve the same required power while wood bark consumption, is the lowest due to the higher heating value considering its elemental composition; see Equation (4).

A set of results shown in Figure 6 at a particular chosen value of the moisture content and $\Phi$ provides the range of the syngas chemical species composition (vol.%) for the various agriculture feedstocks. For fixed working parameters (i.e., 10% MC, and 0.35 $\Phi$), spent hops and spent barley had the highest CO composition at 21.84 and 20 vol.%, respectively. This is because of higher C content in such feedstocks that leads to higher CO production through the Boudouard reaction, which consumes $CO_2$, as illustrated in the same figure. $H_2$, on the other hand, is found in large amounts for both barley dust and barley screenings, where its concentration varies between 18.3% and 21%. Furthermore, compared to the typical gasification process of biomass, the values seem comparable and similar to biomass feedstocks (e.g., wood chips). It is worth analyzing the process in terms of the syngas heating value and the presence of particulates (i.e., tar) within the produced gas (Figure 7). The model assumes the tar as a single compound ($C_6H_{6.2}O_{0.2}$) as considered by many researchers, e.g., refs. [4,34].

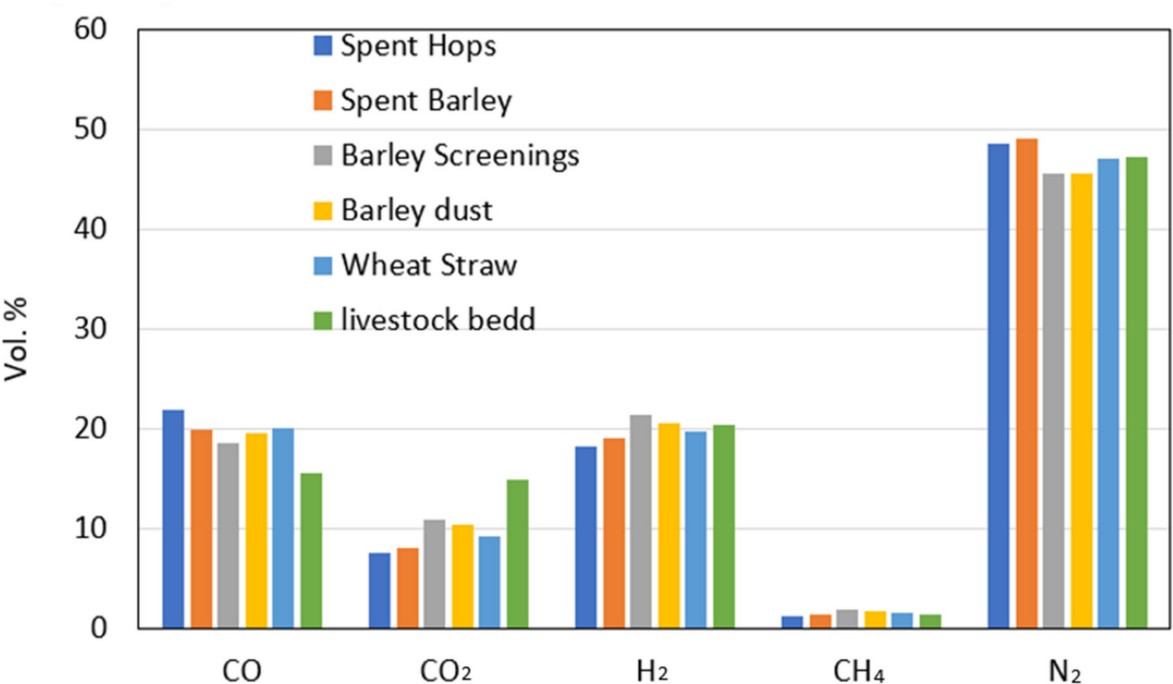

**Figure 6.** Producer gas composition (vol.%) for the different feedstocks at $\Phi = 0.35$ and MC = 10%.

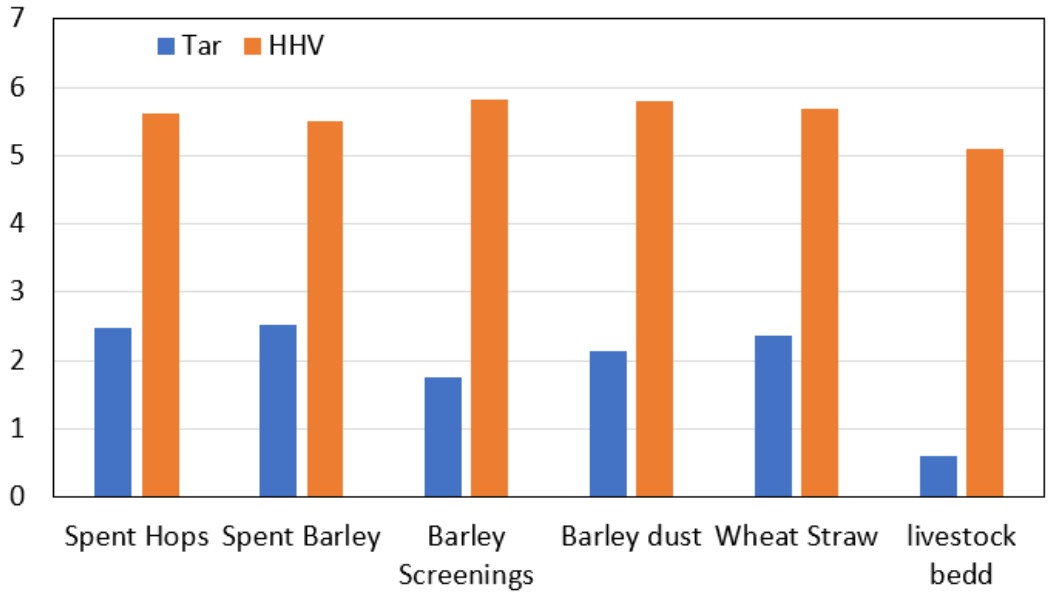

**Figure 7.** Tar (in mol.% of produced gas) and *HHV* (in MJ/Nm$^3$).

Figure 7 shows higher heating values for barley dust and barley screenings (~5.8 MJ/Nm$^3$) with a low tar content of around 2% from the producer gas mole content. This is due to the relatively higher amount of C and H presented in these two feedstocks that enhanced the production of producer gas composition. Further, tar conversion is another important factor that resulted in the production of light compounds resulting in tar reduction. On the other hand, the lowest heating value is found for livestock beddings (5.1 MJ/Nm$^3$), which have the lowest amount of carbon with a high ash content (Figure 4). Lower carbon content in a feedstock tends to decrease the corresponding char oxidation reactions. As a result, it decreases the amount of CO and CH$_4$ produced and hence lowers the heating values. The highest tar levels are found in spent barley, followed by spent hops (2.53% and 2.49%, respectively). This is because their lower oxygen content tends to

decrease the oxidation reactions, resulting in a lower temperature in combustion, lower burning and cracking rates for tars, and hence, higher amounts of tar.

Agricultural feedstocks showed close results to conventional biomass (*HHV* and tar %) as discussed in Figures 6 and 7 because of similar content of CHO and MC levels. Consequently, such feedstocks are economically and technically viable for gasification and syngas production, besides reducing the waste levels.

### 4.2. Sensitivity Analysis

Changing working conditions ($\Phi$ and MC) of feedstocks are further studied to optimize the gasification process and syngas production rate represented in tar content and *HHV*.

Figure 8 illustrates the effect of changing $\Phi$ on producer gas *HHV* while keeping fixed MC at 10%, as recommended by many previous studies for better performance [7,33], while $\Phi$ is changed between 0.2 and 0.4, as previously advised by refs. [4,32,35]. A lower equivalence ratio tends to produce higher amounts of CO, $CH_4$, and $H_2$. As a result, higher heating values are achieved. During gasification, the amount of air injected (which shows the $\Phi$) controls the whole process. Lower air amounts reduce the values of $N_2$ and $CO_2$, leading to a decrease in the dilution of syngas and hence higher syngas content and higher *HHV*. While higher air amounts (higher $\Phi$) lead to an increase in the turbulence, combustion reactions rates result in more consumption of CO, char $CH_4$, and $H_2$ and higher production of $H_2O$, $CO_2$, and $N_2$, which in turn reduce the *HHV* of the produced gas. The results are in good agreement with previous works refs. [12,15,16].

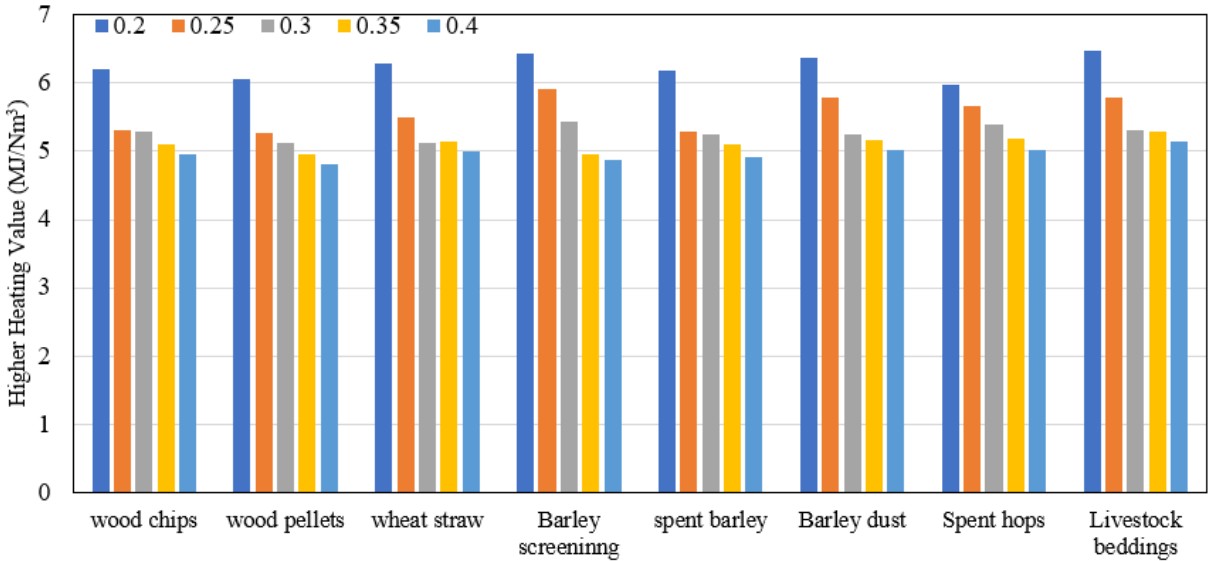

**Figure 8.** Effect of changing equivalence ratio (0.2–0.4) on the *HHV* of the producer gas.

An increase in the producer gas *HHV* is found by 25–30%, while decreasing the equivalence ratio is between 0.4 and 0.2. The results are consistent and match with previous research works (e.g., [13–15]). The highest value of *HHV* was found to be at 6–6.4 MJ/NM$^3$ at $\Phi$ of 0.2, while at $\Phi$ = 0.4, the lowest values of heating value were found to be at 4.8–5.14 MJ/NM$^3$.

Similar values for the *HHV* for most of the feedstocks were found. However, the barley dust, barley screenings, and livestock bedding prove to have the highest values (~6.4 MJ/NM$^3$). Based on their ultimate analysis, such feedstocks require lower air amounts to achieve the same $\Phi$ compared to other feedstocks. As a result, they produce more values of CO, $H_2$, and $CH_4$ (Figure 6) and raise expectations of heating value. However, as previously noted, all materials exhibit the same trend and very close values to heating value production, which is comparable to woody biomass materials (e.g., wood chips). The

results, therefore, demonstrate that using agricultural waste materials in gasification gives potential results.

Furthermore, Figure 9 shows the effect of changing $\Phi$ on producer gas quality (including prediction of tar amounts). Fixed MC levels of 10% were used while varying the $\Phi$. Higher $\Phi$ (air amounts) leads to more turbulence, higher temperatures, and higher oxidation reactions rates [34,36]. Consequently, this increases tar cracking, converts it to lighter compounds, and reduces tar amount in producer gas. For wood pellets, while increasing the equivalence ratio from 0.2 to 0.4, the tar concentration decreases rapidly by 50%, which find a strong agreement with [4,35]. However, as seen in the figure, while increasing the equivalence ratio between 0.2 and 0.4, the tar concentrations fall between 29 and 51% for the selected materials.

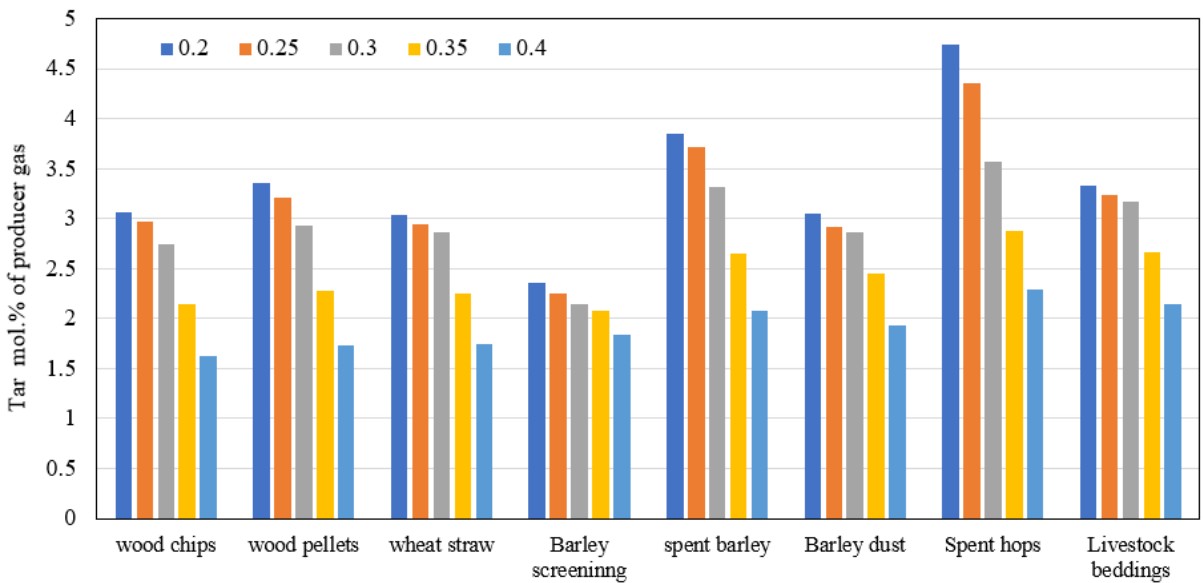

**Figure 9.** Effect of changing equivalence ratio (0.2–0.4) on tar content.

The effect of changing MC on producer gas quality (heating value and tar content) is addressed in Figures 10 and 11. The simulations are run at a fixed ($\Phi = 0.25$) while varying the MC values between 0 and 20%. Higher levels of MC decrease the corresponding heating values, as clearly shown in Figure 10, resulting in lower gasification efficiency. The higher levels of MC in a feedstock will require more energy (heat) for removal, thus affecting the oxidation and gasification reactions. This accordingly dilutes the amounts of syngas yield because of higher moisture content and, as a result, lower the heating value. The results indicate that decreasing MC from 20 to 0% leads to an increase in *HHV* by 1–4% for the selected materials. Barley screenings, barley dust, and livestock beddings thus produced more concentrated syngas and correspondingly higher heating values.

The effect of changing moisture content on the tar produced is illustrated in Figure 11. Moisture content in the feedstock requires more energy and heat for removal. Therefore, the higher MC in a feedstock decreases temperature across the gasifier, leading to lower tar cracking and, as a result, higher tar content. The results reveal a decrease in tar content by 5–7% when the MC decreases from 20 to 0%. Generally, it was found that the *HHV* rises from 25 to 30% when the $\Phi$ decreases from 0.4 to 0.2. Simultaneously, a decrease in MC of biomass from 20 to 0% leads to an increase in *HHV* by 1–4%. As a result, the current study recommends the use of biomass with $\Phi$ of 0.3–0.35 and MC of less than 10% to achieve better performance and concentration of syngas. Additionally, lower $\Phi$ (<0.3) tends to increase the tar amounts in producer gas, as shown in Figure 9.

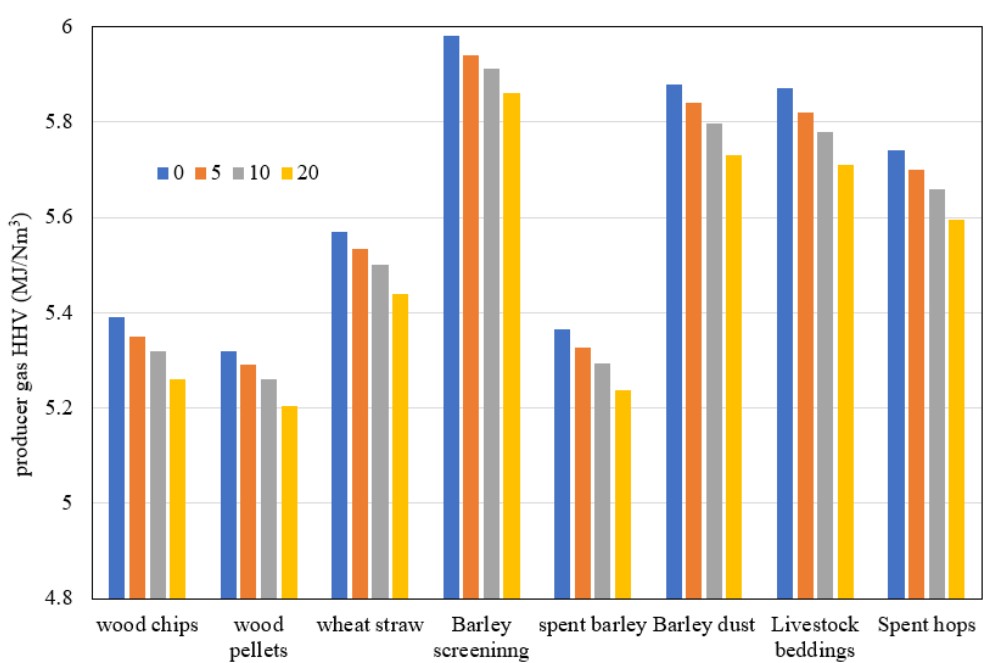

**Figure 10.** Effect of moisture content change (0–20%) on *HHV* of produced gas.

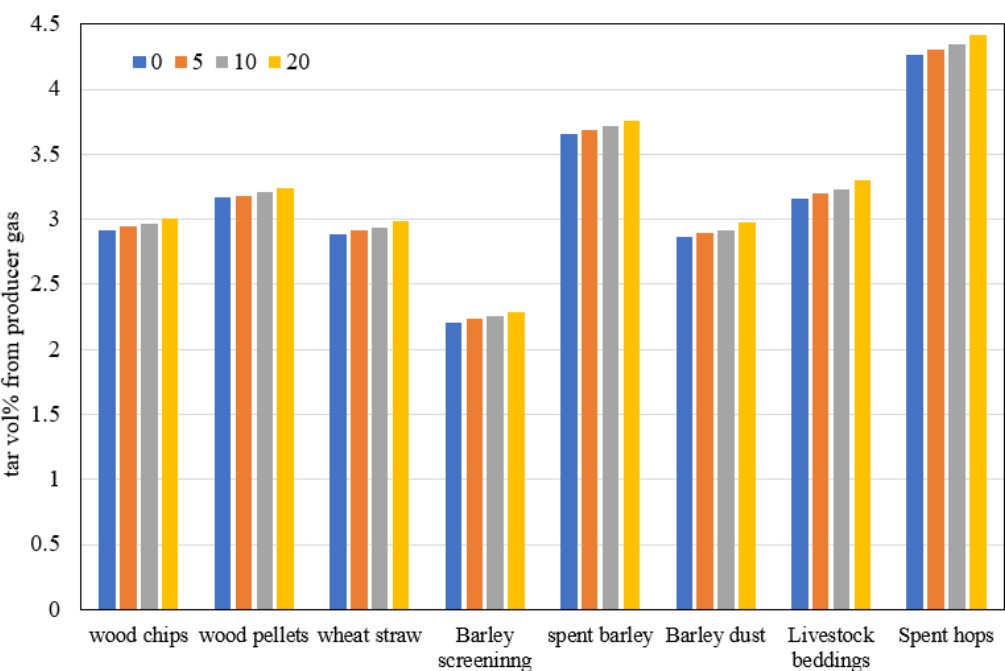

**Figure 11.** Effect of moisture content (0–20%) on tar produced.

### 4.3. Gasification Efficiency

Gasification efficiency is calculated as follows [37]:

$$\eta_{th} = \frac{G_p \, Q_g}{Q_b},\tag{6}$$

where $Q_g$ is the syngas *LHV* (MJ/Nm$^3$), $G_p$ is the yield of producer gas in Nm$^3$/kg, and $Q_b$ is the fuel *LHV* in MJ/kg. The input thermal power is calculated from the *LHV* of biomass along with the biomass feeding rate.

To validate the current model, Figure 12 compares the current model with other experimental results [35] for the gasifier efficiency. Same working conditions are used in the comparison, e.g., corn as feedstock (Table 3), equivalence ratio of 0.24–0.41, and MC (10%).

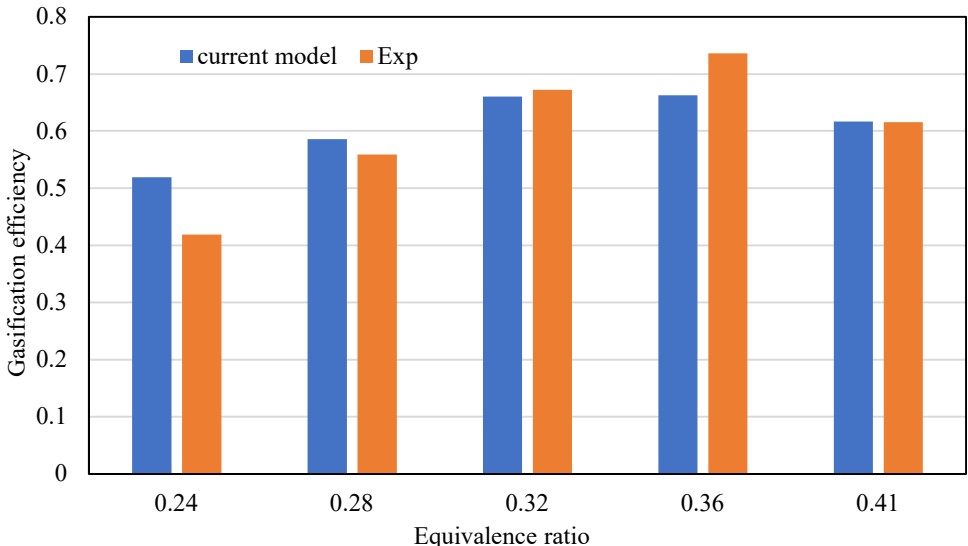

**Figure 12.** Gasification efficiency comparison for the experiments [37] and current model.

**Table 3.** Corn straw elemental analysis.

| Proximate (wt.% db) | | Ultimate Analysis (db. %) | |
| --- | --- | --- | --- |
| Fixed carbon | 13.75 | C | 43.83 |
| Volatile matter | 75.95 | H | 5.95 |
| MC% (as received) | 6.17 | O | 45.01 |
| Ash | 5.93 | N | 0.97 |
| *LHV* (MJ/kg), daf | 17.75 | S | 0.13 |

The same working conditions for comparison and the predicted gasification efficiency were found in good agreement with that of the experimental results at the various $\Phi$s [35] (Figure 12). Following this satisfactory validation, the gasification efficiency for the different feedstocks with an impact of the $\Phi$ is presented in Figure 13.

In Figure 13, a fixed MC of 10% is used while varying the $\Phi$. The highest gasification efficiency was found for wood chips (~66%), especially with the lower value of $\Phi$ (0.2). That is because lower $\Phi$ (lower amount of air) is leading to higher *HHV* and higher concentrations of syngas in the producer gas, which in turn increases the gasification efficiency (Figures 8 and 9). Additionally, wood chips have the highest volatiles and C content, which yields higher CO and $H_2$ content. Furthermore, for the previous discussions regarding higher $\Phi$, it tends to produce syngas that is more dilute (lower heating values), which thus reduces the gasification efficiency.

In general, wood biomass presents higher efficiency for gasification (~66%) because of its more concentrated syngas. This is because of higher volatiles content and CHO levels and reasonable MC values (Figure 4 and Table 1). Again, livestock beddings spent barley and barley screenings have a good gasification efficiency, which is quite like wood materials. On the other hand, lower gasification efficiency was found for spent hops (~38%), with higher tar content (Figure 7), making this feedstock inefficient for gasification, especially when the required drying to obtain lower MC is considered.

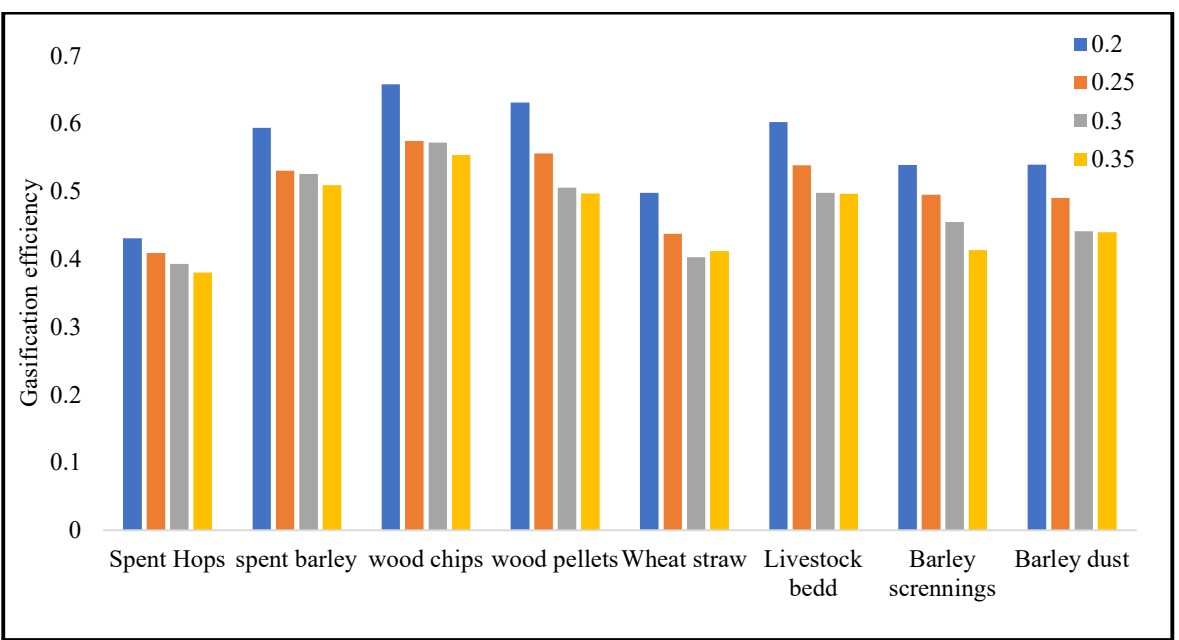

**Figure 13.** Effect of changing $\Phi$ (0.2–0.35) on gasifier efficiency for different feedstocks.

### 4.4. Exergy Analysis

Exergy is the work potential of energy to optimize the efficiency of the process and also to analyze the degree of thermodynamic perfection of an energy system based on the second law of thermodynamics [38,39]. The exergy efficiency is represented by:

$$\eta_{ex} = \frac{\mathrm{E}_{Prod}}{\mathrm{E}_{feed} + \mathrm{E}_{med}} \tag{7}$$

where the *E* refers to exergy; *Prod* refers to the products, i.e., producer gas; *feed* is the feeding, i.e., feedstock; and *med* is the gasifying medium which is neglected when using air. Producer gas exergy is the sum of chemical (*ch*) and physical (*ph*) exergies.

$$\mathrm{E}_{Prod} = \mathrm{E}_{ph} + \mathrm{E}_{ch}. \tag{8}$$

The feeding exergy is estimated from

$$\mathrm{E}_{feed} = \varphi_{dry}[LHV + m_w \, h_{fg}] \tag{9}$$

where the *LHV* is the feedstock lower heating value, $m_w$ is the moisture amount, $h_{fg}$ is the heat removal, and $\varphi_{dry}$ is estimated from the biomass ultimate analysis *CHON* as follows:

$$\varphi_{dry} = \frac{1.049 + 0.016\frac{H}{C} - 0.3493\frac{O}{C}\left(1 + 0.7256\frac{H}{C}\right) + 0.0493\frac{N}{C}}{1 - 0.4124\frac{O}{C}}. \tag{10}$$

A comparison between the current model exergy analysis and the previous work of [38] is illustrated in Figure 14. The results are carried out for the same feedstock (solid residues, mixture of MSW) for different MC levels, as shown in the figure. The results demonstrate a fairly good agreement between the current model and previous work with negligible variations.

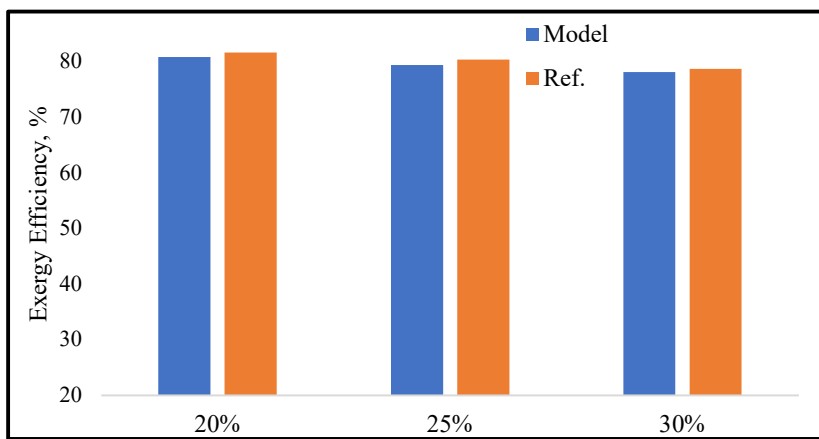

**Figure 14.** Second law efficiency comparison for present and previous work of [38].

The data demonstrated in Figure 15 show different exergy efficiencies for the feedstocks of the current study. Compared to wood biomass materials, the agricultural feedstocks show similar results as previously discussed for the *HHV* and producer gas composition. Such similarity was expected because of the similar analysis and composition of the selected materials. The second law analysis was estimated at fixed *Φ* of 0.3, while MC = 10% as suggested for better performance of the gasification process, as previously illustrated in Section 4.2. The highest value for exergy efficiency was noted for livestock beddings (~87%) because of the higher syngas yield.

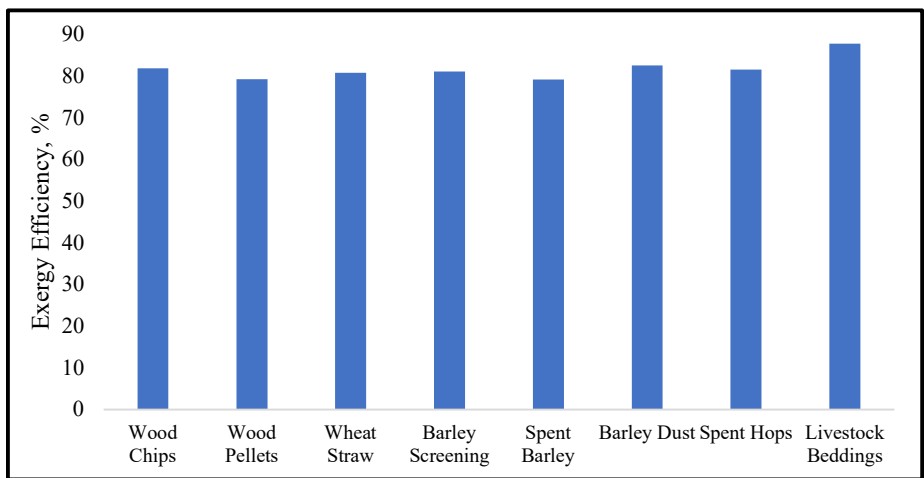

**Figure 15.** Exergy efficiency for the agricultural residues.

### 4.5. Gasifier Design Limitations and Optimisation

It is of great importance to investigate the impact of feedstocks' chemical composition on the design of a gasifier for a required syngas production rate and thermal power. The variety of feedstocks in the current research besides the biomass materials that are normally used in gasification makes the optimization process of a gasifier design very complicated. A gasifier design depends on specific variables, e.g., thermal power required, the feedstock, and working conditions. A gasifier design (dimension) is illustrated in detail at [4], in which the fuel feeding rate is calculated from the required thermal power and feedstock heating value. Then, based on the feeding rate, the throat diameter is calculated. Other gasifier dimensions, e.g., pyrolysis diameter and height, are all derived from the throat diameter. However, as discussed earlier, the feedstocks variety makes a significant challenge when using the same design for a gasifier because it might affect the gasification process. These key challenges are addressed in this section.

For a 20 kW gasifier design, the set of results presented in Figure 16 shows the main gasifier dimensions, the corresponding feeding rate, and the heating value of producer gas. Except for the length of the reduction zone, which varies between 29 and 42 cm, the results indicate homogeneity in other dimensions. However, the designed kinetic code considers the optimum height of the reduction length/height for a full char consumption. Consequently, increasing the height of reduction has no effect on syngas production or the gasifier power. Additionally, a slight/negligible variation for the pyrolysis/firebox diameter, throat, and oxidation area height was found.

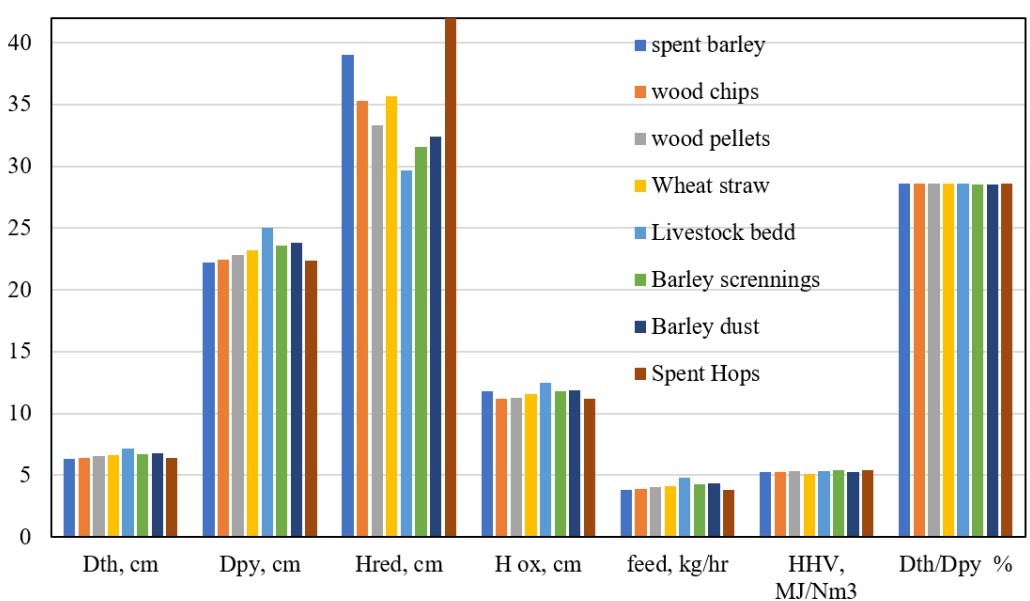

**Figure 16.** Gasifier design (various dimensions) for different feedstocks/materials.

The results reveal a key design dimensionless parameter—the ratio of the throat to gasifier ($D_{th}/D_{py}$) diameter—which affects the syngas production and the corresponding power. An optimum value for all the feedstocks (for $D_{th}/D_{py}$) of around 29% was found, which met an agreement with previous experiments [40,41].

A wide range of CHO for the materials was used in the current simulations. It was found that designing a gasifier based on spent barley and spent hops suits most of the other agricultural feedstocks. The higher CH content in such feedstocks releases more C (char) in the combustion and reduction zones. As a result, it affects reduction length, which in turn affects the gasification reactions and char consumption, thus requiring a higher length for gasification/reduction to accommodate char formation and consumption and the corresponding reactions. The selected feedstocks have moderate dimensions for the gasifier design based on a fixed power (20 kW). To further prove this point, the reversed calculations (Figure 17) are introduced to assess and study the effect of change of heating value and syngas production.

On the other hand, they have a moderate value for the other design dimensions of gasifiers based on the fixed power rate. To prove this point, the optimal producer gas *HHV* for the different feedstocks in Figure 16, calculated using the kinetic model, was compared to the producer gas *HHV* that would be produced by these feedstocks if the gasifier was designed using the dimensions and parameters for spent barley (Figure 17).

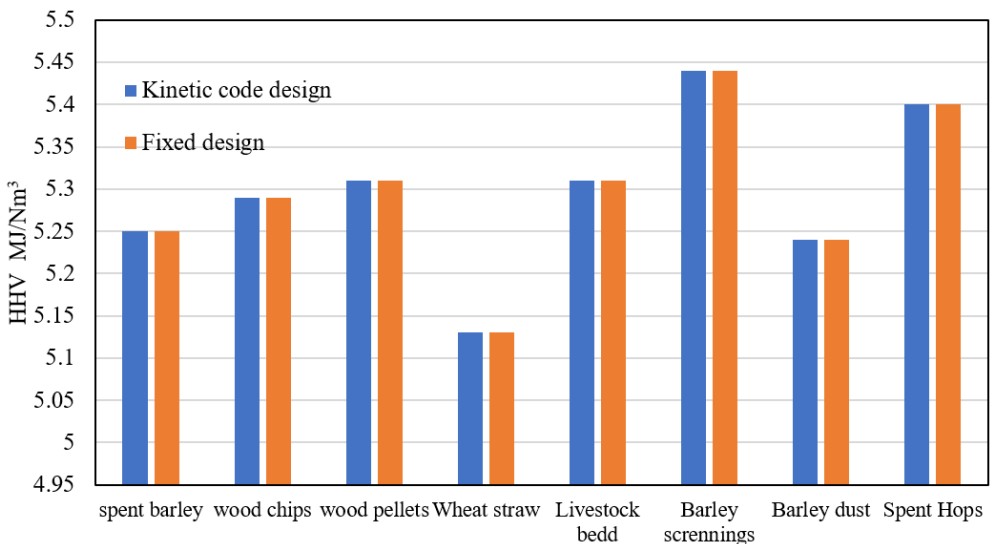

**Figure 17.** Effect of using fixed design on the *HHV* for different feedstocks.

In Figure 17, this model is performing the effect of using a fixed thermal power and fuel feeding rate for spent barley feedstock (specific gasifier design). A fixed gasifier dimension ensures the same fuel consumption for all of the feedstocks, and hence, the biomass feeding rate is assumed to be constant with the thermal input power. Thermal power of 20 kW and fixed feeding rate of 3.79 kg/h (same for spent Barley) are used in the model for $\Phi = 0.3$ and MC of 0.1. The results in Figure 17 for the heating value have no difference in *HHV* when using the same design of a gasifier. Therefore, constructing a gasifier based on the highest C and H contents of a specific feedstock works properly with other feedstocks and gives the expected production of syngas. This is due to the required specific dimensions (higher reduction length) to ensure all the char consumption within the gasification/reduction zone, resulting in the ultimate production of syngas and hence the highest heating value.

## 5. Conclusions

A detailed model was built to combine gasification with CHP engines for power generation, as well as to determine working parameters to optimize the process. Agricultural waste was used as a fed into the gasifier, and the producer gas was combusted to produce power using an ICE. In addition to the gasifier design dimensions, gas concentrations, and tar content, the model was able to determine the feed rates required to deliver the electrical and thermal outputs from the CHP engine at any given load. The model was designed based on the chosen agriculture feedstocks; however, it was further used to simulate other biomass materials and found good agreement with reasonable predictions.

Moreover, sensitivity analysis was performed by investigating the effect of air equivalence ratio and moisture content levels on produced gas quality, which affected the power generated from the CHP engine. The highest heating value was found for barley dust and barley screenings (~6.4 MJ/NM$^3$) with moderate tar content levels (~2% mol.% of produced gas). The results of the model recommend the gasification under $\Phi$ of 0.3–0.35 and lower MC levels of <10% to achieve higher concentrations of syngas produce, *HHV*, gasification efficiency, and lower tar amounts.

Furthermore, the model investigates the efficiency of gasification, and the model was used to address design limitations/challenges associated with utilizing different feedstocks. The final results detailed the process of optimizing a gasifier design that can handle and operate under different working conditions for diverse feedstocks while still be producing higher quality syngas.

**Author Contributions:** Conceptualization, A.M.S. and M.C.P. methodology, A.M.S., H.S.D. and M.C.P.; software, A.M.S. and H.S.D.; validation, A.M.S. and H.S.D., resources, A.M.S. and H.S.D.; writing—original draft preparation, A.M.S. and H.S.D.; writing—review and editing, A.M.S. and M.C.P.; supervision, M.C.P. All authors have read and agreed to the published version of the manuscript.

**Funding:** This research was funded by the Scottish Government Interface Food & Drink (IFD0190) and the University of Glasgow KE Fund (GKE100).

**Institutional Review Board Statement:** Not applicable.

**Informed Consent Statement:** Not applicable.

**Data Availability Statement:** Not applicable.

**Acknowledgments:** We gratefully acknowledge the Scottish Government Interface Food & Drink (IFD0190) and the University of Glasgow KE Fund (GKE100) for the financial support. M.C.P. also acknowledges his RAEng/Leverhulme Trust Senior Research Fellowship support (LTSRF1718\14\45) from the Royal Academy of Engineering, UK. The research novelty and aims are based on the first author's Ph.D. Thesis available at the University of Glasgow's repository [42].

**Conflicts of Interest:** The authors declare no conflict of interest.

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
