# Peer review of "Syngas Production and Combined Heat and Power from Scottish Agricultural Waste Gasification—A Computational Study"

_sustainability, doi:10.3390/su14073745_

Round 1

Reviewer 1 Report

Biomass gasification processes are known and widely described in many publications. Hence, the presented material shows a high absolute similarity index of 18%, excluding bibliographies and citations. Most of the similarities concern terminology and definitions, which may be acceptable, but 8% of the similarities concern the doctoral dissertation of one of the Authors, i.e. Ahmed M. Salem, entitled: "Investigation of biomass gasification processe for the production of high quality syngas" realized at the College of Science and Engineering, University of Glasgow in 2019. This doctoral dissertation was not found in the bibliography. For this reason, I have significant reservations about the originality of the material sent. Regardless of this, I also have reservations about the bibliography used by the Authors. The scientific publications used by the Authors are outdated. The only item comes from 2021 (item 15) and the others are from earlier years, including even 2004 (item 20). Gasification processes are the subject of research in many research centers. There are a number of current publications in this regard. The only positive aspect of the presented material is the attempt to optimize the design of gasifiers. This possible optimization should, however, be confirmed by research on the efficiency of the gasification process for mixtures of specific biomass raw materials, not only from a specific cultivation area. Summing up, I believe that the submitted material does not meet the requirements for original scientific material enabling its publication in "Sustainability". This material should be rejected.

I propose the authors to prepare a new article containing:

  1. The current analysis of the state of knowledge in the field of the possibility of producing high-quality syngas from biomass with identified areas of its possible use.
  2.  Development of the gasifier design and possibly its prototype enabling the production of synthesis gas of a specific composition.
  3. Optimization of the gasification process for mixtures of biomass raw materials of an identified type, origin and proportions of the share of individual components of these mixtures.
  4. Presentation of unambiguous conclusions that can be used in further research and in industrial practice.

Reviewer 2 Report

  • The article is interesting and considerable for publication

  • All the study work has been carried out in Simulink/Simulation-based. It should be validated with physical model development if possible

  • In methodology, the author says that gas produced is not sufficient for CHP

  • In optimum gasifier design from results tells that it is very difficult to design, therefore using the same design will be a challenge that will ultimately affect the gasifier performance.

  • Moisture content of almost all the crops are more than 10%, how this can be reduced to less than 10% for better performance of the gasifier. Also load rate or feeding rate is very important. How feeding rate can be managed.

  • Economics factor is missing as it plays a key role in the success or failure of an idea therefore that must be incorporated for better understanding of real profit.

  • There looks no potential of agricultural waste as raw material availability, therefore if the raw material is not available then this study does not look feasible, however, kitchen/urban waste has decent potential if considered. Though agricultural waste is available but in the introduction part, the author says that it is either used and feed for animals or please clear it.

Reviewer 3 Report

General comments: The submitted manuscript examines the possibility of utilising Scottish agricultural waste for sustainable energy, including combined heat and power (CHP). The content of the paper is in line with the journal's scope and is suitable for the readership of the journal. I am grateful to the authors for their work and effort in preparing the paper. However, I think there is a scope for improvement. More comments can be found in the specific comments section.

Specific comments:

  • Originality/Novelty: The system is not novel. I would suggest improving the methodology. It will improve the quality of the work.
  • I would suggest improving the quality of the presentation.
  • Scientific Soundness Technically, the paper is OK. It has scope for improvement. However, better methodologies should be incorporated.
  • Rewrite the introduction. Try to establish the research gap and specify why this study is important.
  • Provide a separate table that shows the mass flow rate, temperature, pressure, specific enthalpy, specific exergy, and stream composition at various state points in Figure 1.
  • What are the assumptions employed in the analysis? It should be provided.
  • All the major input parameters should be given in a table.
  • As this system is a CHP system, exergy efficiency of the plant should be calculated.
  • Experimental results must be used to compare with the syngas composition of the gasifier model.
  • Include economic analysis and multiobjective optimization to improve the content of the work.
  • What was the chemical formula of tar considered in your simulation?
  • The ranges selected for the operating parameters should be properly justified.
  • Focus more on recent papers published in the last 5 years. Older references may be omitted. You may add the following recent papers to the introduction:

https://doi.org/10.1016/j.energy.2020.117268

https://doi.org/10.1016/j.biombioe.2022.106370

https://doi.org/10.1016/j.enconman.2020.113182

https://doi.org/10.1016/j.seta.2020.100867

  • Please keep self-citations to a maximum of 4-5 papers.
  • Compare your findings to those of other similar studies.

Round 2

Reviewer 1 Report

The Authors introduced clarifications and supplements in line with my suggestions. I don't think the Authors have fully understood my intentions. I expected research on the optimization of gasifier operation for specific mixtures of raw materials, regardless of their place of origin, and an analysis of possible areas of use of the obtained result of this process. However, taking into account the introduced corrections and additions, I can consider the presented material useful for further research. Positive comments from other Reviewers are of course important, but they cover other aspects of the presented material and cannot constitute arguments decisive for the full suitability of the presented material for the implementation of biomass gasification processes, which can be obtained not only in Scotland. I believe that the Authors can meet my expectations in their further research activities.

Author Response

This has been replied in the 1st stage.

Reviewer 2 Report

Accept in present form

Author Response

This has been replied previously, and no further comments arised.

Reviewer 3 Report

The revised manuscript is greatly improved and can now be accepted.

Author Response

submitted in the 1st round